# Comparison and Analysis of Neutralizing Antibody Levels in Serum after Inoculating with SARS-CoV-2, MERS-CoV, or SARS-CoV Vaccines in Humans

**DOI:** 10.3390/vaccines9060588

**Published:** 2021-06-02

**Authors:** Sicong Yu, Keda Chen, Lei Fang, Haiyan Mao, Xiuyu Lou, Chaonan Li, Yanjun Zhang

**Affiliations:** 1School of Medical Technology and Information Engineering, Zhejiang Chinese Medical University, Hangzhou 310053, China; yu.sicong@foxmail.com; 2Zhejiang Provincial Center for Disease Control and Prevention, Hangzhou 310051, China; lfang@cdc.zj.cn (L.F.); hymao@cdc.zj.cn (H.M.); xylou@cdc.zj.cn (X.L.); 3Shulan International Medical College, Zhejiang Shuren University, Hangzhou 310015, China; chenkd@zjsru.edu.cn (K.C.); 201930001207@stu.zjsru.edu.cn (C.L.)

**Keywords:** SARS-CoV-2, MERS-CoV, SARS-CoV, vaccine, neutralizing antibody

## Abstract

Severe acute respiratory syndrome coronavirus 2 (SARS-CoV-2), Middle East respiratory syndrome coronavirus (MERS-CoV), and severe acute respiratory syndrome coronavirus (SARS-CoV) pose a great threat to humanity. Every pandemic involving these coronaviruses has seriously affected human health and economic development. Currently, there are no approved therapeutic drugs against their infections. Therefore, the development of vaccines is particularly important to combat these coronaviruses. In this review, we summarized and analyzed the progress of vaccines against SARS-CoV, MERS-CoV, and SARS-CoV-2, including inactivated vaccines, live attenuated vaccines, subunit vaccines, nucleic acid vaccines, and viral vector vaccines. In addition, we compared the levels of neutralizing antibodies in the serum of patients with these three kinds of coronaviruses at different stages, and their ability and effects against SARS-CoV-2, MERS-CoV, and SARS-CoV. This review provides useful information for vaccine evaluation and analysis.

## 1. Introduction

In the past 18 years, three novel coronaviruses—severe acute respiratory syndrome coronavirus 2 (SARS-CoV-2), Middle East respiratory syndrome coronavirus (MERS-CoV), and severe acute respiratory syndrome coronavirus (SARS-CoV)—have jumped the species barrier, infected humans, and caused human-to-human transmission. In addition, they all belong to β-coronavirus. The genomes of SARS-CoV-2, MERS-CoV, and SARS-CoV are positive-sense single-stranded RNA and have a 5′-cap and 3′-UTR poly(A) tail, which is approximately 25–32 kb [1]. The genomes of coronavirus encode four major structural proteins, including spike (S) protein, envelope (E) protein, membrane (M) protein, nucleocapsid (N) protein, and accessory proteins (ORF 3a, 3b, 6, 7a, 7b, 8b, and 9b) (Figure 1A–D) [2]. There is 79% sequence identity between SARS-CoV-2 and SARS-CoV, and 50% sequence identity between SARS-CoV-2 and MERS-CoV [3]. The spike protein consists of S1 and S2 subunits and facilitates the binding of the viral envelope to angiotensin-converting enzyme 2 (ACE2) receptors expressed on many mammalian organs such as the lung, kidney, gastrointestinal tract, and heart [4]. The N protein is involved in RNA replication, virion formation and also in immune evasion [5]. The interaction between M protein, N protein, and accessory proteins 3a and 7a promotes the assembly and budding of viral particles [6]. The function of E protein is to facilitate the production, maturation and release of virions [5].

Current research on coronavirus disease 2019 (COVID-19) suggests that the earliest outbreak began on 1 November 2019 [7]. Shortly thereafter, SARS-CoV-2 was identified as the virus that caused a series of cases of “unexplained pneumonia” in Wuhan, China, in December 2019. As of 15 May 2021, a total of 161,513,458 people is reported to have been infected with SARS-CoV-2, and the death toll has reached 3,352,109 according to the latest data from the World Health Organization (WHO) [8]. According to the WHO, 99 vaccine candidates are in clinical development, and phase III clinical trials of candidate vaccines have been initiated for the adenovirus type 5 (Ad5) vector vaccine (CanSino Biological/biotechnology research institute, Beijing, China, Ad26-based vector vaccine (Janssen/Johnson & Johnson, Titusville, NJ, USA), the chimpanzee adenovirus vector vaccine (Astrazeneca, Cambridge, UK/Oxford University, Oxford, UK), Ad5 and Ad26-based vector vaccine (Gamaleya, Moscow, Russia), inactivated virus vaccine (Sinovac Biotech, Beijing, China; Wuhan Institute of Biological Products, Wuhan, China/Sinopharm, Beijing, China; Beijing Institute of Biological Products, Beijing, China/Sinopharm, Beijing, China; Bharat Biotech, Hyderabad, India), protein subunit-based vaccines (Novavax, Gaithersburg, MD, USA; Anhui Zhifei Longcom Biopharmaceutical, Anhui, China/Institute of Microbiology, Chinese Academy of Sciences, Beijing, China; Sanofi, Paris, France/GSK, Brentford, UK), mRNA vaccines (Moderna, Cambridge, US/NIAID, MA, USA; Pfizer, New York, NY, USA/BioNTech, Mainz, Germany; CureVac AG, Standort Tübingen, Germany), and DNA vaccines (Inovio Pharma, Missouri, MO, USA/International Vaccine Institute, Seoul, Korea).

MERS is a novel coronavirus discovered in 2012. The virus first appeared in Saudi Arabia, and most people infected with MERS develop severe respiratory syndrome and acute kidney failure. The virus can be transmitted from dromedary camels to humans and from human to human. Infection with the virus can lead to respiratory disease and death in up to 35% of cases. As of the end of April 2021, a total of 2574 cases of MERS have been reported in 27 countries worldwide, including 886 (34.4%) associated deaths. At the end of December 2020, Saudi Arabia had the highest number of cases (2167) [9]. To date, there is no effective vaccine against MERS coronavirus, and no specific drug is available to treat it. MERS vaccines currently in clinical trials include vaccines based on the attenuated chimpanzee adenovirus vector Chadox1, the modified vaccinia virus Ankara MVA, and the DNA vaccine GLS-5300.

SARS was a global epidemic that began in Guangdong, China in 2002 and spread to Southeast Asia and then worldwide. It was gradually eliminated in the middle of 2003. The clinical symptoms of patients with mild SARS are light, whereas patients with SARS are prone to respiratory distress syndrome. By July 2003, with a total of 8096 cases reported worldwide [10], including 774 deaths in 27 countries, no further infections were detected, and the SARS pandemic was over.

## 2. Clinical Trials on the Ability of Different Types of SARS-CoV-2 Vaccines to Induce Neutralizing Antibody Levels after Vaccination

SARS-CoV-2 vaccines can elicit a protective immune response, which is essential to prevent and reduce the morbidity and mortality of SARS-CoV-2 infection. Current understanding suggests that a balanced humoral and Th1-directed cellular immune response might be important for protection against COVID-19 and the avoidance of vaccine-enhanced disease [11]. A variety of vaccine candidates are being developed and tested, including nucleic acid vaccines, inactivated virus vaccines, attenuated live vaccines, protein or peptide subunit vaccines, and viral vector vaccines (Figure 1E). Each type of vaccine has its advantages and disadvantages [12]. The leading candidates are delivered via intramuscular injections; therefore, the focus is to assess the immune response in the blood. Several candidate vaccines have been subjected to clinical trials (Table 1). In addition, specific vaccine formulations may be required in different populations, for example, immunologically immature infants, children, pregnant women, immunocompromised persons, and primary immunized persons aged 65 years or older.

### 2.1. SARS-CoV-2 Inactivated Vaccine

Inactivated vaccines have been widely used in the influenza vaccine, and their preparation technology was mature. Therefore, formaldehyde, ultraviolet light, and β-propiolactone have been used to prepare inactivated vaccines against SARS-CoV-2, which usually induce a potent serum neutralizing antibody response. The inactivated vaccine is the most mature vaccine technology; for example, the polio vaccine, hepatitis A vaccine, hand, foot, and mouth vaccine, and rabies vaccine are all inactivated vaccines.

The inactivated vaccine Coronavac, developed by Sinovac Biotech, Beijing, China, was made by inoculating Vero cells with novel Coronavirus after virus culture, harvesting, inactivation, purification, and aluminum adsorption. Results of preclinical trials showed that the vaccine-induced specific SARS-CoV-2 neutralizing antibodies in mice, rats, and non-human primates neutralized 10 representative SARS-CoV-2 strains. The use of 3 μg or 6 μg per dose provided partial or complete protection against SARS-CoV-2 attack in rhesus monkeys, respectively, and no increase in antibody-dependent infection was observed [13]. The safety data of the phase I/II trial showed that the adverse reactions of the vaccine were mainly graded 1, manifesting as mild pain at the inoculation site, fatigue and, low fever in individual subjects, and no serious adverse reactions were reported.

The results of the phase II clinical trial (NCT04352608) showed that the seroconversion rate of neutralizing antibodies was more than 90% after 14 days of whole-course immunization, and the vaccine was safe and well-tolerated at all studied doses. The vaccine is currently undergoing phase III clinical trials in Brazil, Indonesia, and other countries [14]. Coronavac could be an attractive option because it can be stored in a standard refrigerator at 2–8 °C, which is how many existing vaccines, including influenza vaccines, are stored. The vaccine is also stable over a storage period of up to three years, providing an advantage for distribution, especially in areas that lack cold storage. The interim report of the Phase III trial (NCT04651790) showed that the seroconversion rate of neutralizing antibodies was more than 90%, both in the 18–59 age group and the ≥60 age group, but the seroconversion rate of anti-N-SARS-CoV-2 IgG was not detected [15]. Wuhan Institute of Biological Products/Sinopharm is developing and testing an inactivated virus vaccine, and Phase I and Phase II (ChiCTR2000031809) study data have been released [16]. The phase II trial looked at a comparator of the vaccine administered on days 0 and 14 and days 0 and 21, excluding convalescent patient plasma. Only the neutralizing antibody titer was measured, and the concentration was similar to that produced by other COVID-19 vaccines, being higher in the group 0 and 21 days after vaccination (GMT247, 95% confidence interval (CI) 176–345) than in the group 0 and 14 days after vaccination (GMT121, 95% CI 95–154). The seroconversion rate of neutralizing antibodies was 97.6% after two doses inoculated in 0–14 days and 0–21 days, and 100% after two doses inoculated in 0–28 days. Phase III clinical trials begin in July 2020, and they plan to enroll 21,000 participants in countries such as the United Arab Emirates.

The inactivated vaccine (BBIBP-CorV), developed by the Beijing Institute of Biological Products, Beijing, China/Sinopharm, Beijing, China, was immunized with two doses in phase I/II clinical trials (ChiCTR2000032459). The vaccine was safe and well-tolerated at all two doses in both age groups, and rapid humoral response to SARS-CoV-2 was observed beginning on day 4 after the initial vaccination, and 100% seroconversion was observed in all participants on day 42 [17]. In preclinical studies, immunization with BBIBP-CorV induced high neutralizing antibody titers against SARS-CoV-2 in mice, rats, guinea pigs, rabbits, and non-human primates (cynomolgus monkeys and macaques) [18].

The inactivated vaccine (BBV152) developed by Bharat Biotech, hyderabad, India, is a whole-virion inactivated SARS-CoV-2 vaccine formulated with a toll-like receptor (TLR) 7/8 agonist molecule adsorbed to alum (Algel-IMDG). This vaccine can be stored and transported at 2–8 °C. The phase I clinical trial (NCT04471519) showed that the vaccine could induce cellular and humoral immunity. There was no significant difference in the GMT on the 42nd and 104th days among the vaccine groups [19]; this vaccine has shown the possibility of durable cellular immunity and humoral immunity. In the Phase II trial (NCT04471519), participants were divided into the 3 μg group and the 6 μg group. Seroconversion based on PRNT IC_50_ was more than 90% in all dose groups, and the GMT of NAb in the 6 μg group was higher than that in the 3 μg group. It is irrational that the phase II trial was not set up as the control group; this was because they wanted more participants in the experimental group [20]. The phase III trial (NCT04641481) will test the efficacy of 6 µg with Algel-IMDG. On day 56, the serum collected from participants in the 6 µg with Algel-IMDG group had equivalent NAb titers (PRNT IC_50_) to the UK-variant (hCoV-19/India/20203522, B.1.1.7 or 20B/501Y.V1) and heterologous strain (hCoV27 19/India/2020Q111) [21]. In other words, BBV152 is still effective against the UK variant (B.1.1.7 or 20B/501Y.V1).

### 2.2. SARS-CoV-2 Vector Vaccines

The chimpanzee adenovirus vector research vaccine (ChAdOx1 nCoV-19), developed by Oxford University, Oxford, UK and AstraZeneca, Cambridge, UK, encodes the spike protein of SARS-CoV-2. Adenovirus vector vaccines have the advantage of being able to produce more persistent cellular immunity. ChAdOx1 nCoV-19 uses a chimpanzee adenovirus vector. In preclinical studies, the vaccine showed both immunogenicity and protective efficacy in non-human primates under primary immune-booster vaccination regimens [22]. The results of phase I/II clinical trials (NCT04324606) showed that most adverse reactions after injection of the vaccine were mild. A single dose induced humoral and cellular reactions against SARS-CoV-2. A booster dose of the vaccine could increase the titer of the neutralizing antibody [23]. As of 6 April 2021, 32 million adults have been vaccinated in the UK and over 5 million have had both doses. However, some serious adverse events were reported during the use of the vaccine, referred to as vaccine-induced immune thrombotic thrombocytopenia. The five cases [24] (four of the patients had a major cerebral hemorrhage) occurred in a population of more than 130,000 vaccinated persons, and 23 patients [25] presented with thrombosis and thrombocytopenia after receiving the first dose of the vaccine. Most of the patients (except one patient out of the 23 patients) had high levels of antibodies to platelet factor 4 (PF4).

The recombinant adenovirus serotype 5 (Ad5-nCoV) vector vaccine was developed to express the full-length spike protein of Wuhan-Hu-1 virus strain SARS-CoV-2, which was developed by CanSino Biological, Beijing, China/Beijing Institute of Biotechnology, Beijing, China. Ad5-nCoV uses a recombinant, replication-incompetent human adenovirus type 5 (Ad5) vector. The vaccine was approved for military use in China on 25 June 2020, which is the first limited-use COVID-19 vaccine to be approved. The phase I clinical trial (NCT04313127) tested three doses, 5.0 × 10^10^ viral particles, 1.0 × 10^11^ virus particles, and 1.5 × 10^11^ virus particles, respectively. Specific humoral responses against SARS-CoV-2 peaked at day 28 post-vaccination, and rapid, specific T-cell responses were noted from day 14 after one shot of the vaccine. Adverse reactions included pain, redness, and swelling at the injection site, fever, headache, fatigue, and muscle or joint pain, most of which were mild [26]. A phase II clinical trial (NCT04341389) was used to test the two doses, 5.0 × 10^10^ viral particles, and 1.0 × 10^11^ viral particles. On day 28, 148 participants (59%) in the high-dose group and 61 participants (47%) in the low-dose group showed seroconversion of the neutralizing antibody responses to live SARS-CoV-2. On day 28, 244 participants (96%) in the high-dose group and 129 participants (97%) in the low-dose group showed seroconversion to RBD-specific ELISA antibodies [27]. Phase III clinical trials of the vaccine will be conducted in Saudi Arabia and Russia.

The Ad26.COV2.S is a recombinant, replication-incompetent adenovirus vaccine, developed by Janssen/Johnson & Johnson, NJ, USA, encoding a full-length and stabilized SARS-CoV-2 spike protein. The Ad26.COV2.S uses a recombinant, replication-incompetent human adenovirus type 26 (Ad26) vector; these vaccines can be manufactured in mammalian cell lines at scale, providing an effective, flexible system for high-yield manufacturing—other Ad26-based vaccines include an Ebola vaccine [28]. In the phase I/IIa trial (NCT04436276), participants were divided into the low-dose group (5 × 10^10^ viral particles) and the high-dose group (1 × 10^11^ viral particles) in a single-dose or two-dose schedule. In 90% of the participants on day 29 after the first vaccine dose, neutralizing-antibody titers were 224–354. A second dose provided an increase in the titer by a factor of 2.6 to 2.9 [29]. In one-dose regimens, neutralizing antibody responses were stable for at least 14 weeks. Neutralizing antibody titers against the variants containing the E484K substitution were present but reduced 3.35–7.78-fold, and the B.1.351 spike variant was neutralized with a 5.02-fold reduced titer [30]. In the phase III trial (NCT04505722), the result has shown that the vaccine efficacy of one dose of Ad26.COV2.S (5 × 10^10^ viral particles) was 66.9% [31]. Participants had cerebral venous sinus thrombosis (CVST) with thrombocytopenia, thrombosis with thrombocytopenia syndrome (TTS), and positive results of heparin-induced thrombocytopenia (HIT) antibody test [32]. These serious adverse events that occurred in women aged 18–49 led to a pause in the use of the Janssen COVID-19 vaccine for a period of time in April 2020. As of the most recent follow-up, three patients had died, while four remained in an ICU [33]. The vector vaccine (Sputnik V) developed by Gamaleya from Russia consists of two doses, the first of which is a recombinant adenovirus serotype 26 (rAd26) vector and the second of which is a recombinant adenovirus serotype 5 (rAd5) vector, both of which carry genes for the SARS-CoV-2 spike protein (rAd26-S and rAd5-S). Results from the phase I/II clinical trial (NCT04436471, NCT04437875) showed that 100% of the recipients had undergone seroconversion, with RBD ELISA titers and neutralizing antibody titers equal to or greater than those observed in the plasma of COVID-19 convalescent patients [34]. The final data from the phase III clinical trial showed that the vaccine was 91.4% effective overall, and 100% effective in protecting against severe cases of COVID-19 [35].

### 2.3. SARS-CoV-2 DNA Vaccines

A DNA vaccine (INO-4800) developed by Inovio Pharmaceuticals, Missouri, USA is a candidate vaccine targeting the SARS-CoV-2 spike protein. Inovio has extensive experience in developing coronavirus vaccines and is the only company to have a MERS coronavirus vaccine in phase IIa trials. Following the release of the novel coronavirus gene sequence, Inovio used its proprietary DNA drug platform to rapidly design INO-4800. In the study of SARS-CoV-2 virus attack, robust neutralizing antibodies and T-cell immune responses in multiple animal models support the ongoing INO-4800 clinical trial [36]. In the phase I clinical trial (NCT04336410), 94% of participants produced the expected immune response, including neutralizing antibodies and T-cell immune responses, with no serious adverse reactions [37]. In November 2020, the vaccine will enter phase II/III clinical trials (NCT04642638).

A DNA vaccine (GX-19), developed by South Korea’s Genexine, has been studied in primates and has been demonstrated to produce antibodies that neutralize the novel coronavirus. The phase I/IIa clinical trial (NCT04445389) began in June 2020, and the estimated study completion date is June 2022 [38]. Another form of DNA vaccine that is at the stage of clinical trials targets the spike protein of SARS-CoV-2 and was jointly developed by Osaka University/ANGES/Takara Bio/Cytiva/Brickell Biotech. At present, it has been tested on animals and started a clinical trial on 30 June 2020. The vaccine is mainly manufactured from Escherichia coli, and the raw material is easy to obtain so that it can be prepared in a short time for use in emergencies such as COVID-19. The antibodies that are produced by injecting the vaccine into horses could be used as a therapeutic drug for seriously ill patients.

### 2.4. SARS-CoV-2 mRNA Vaccines

With high efficiency, safety, low production cost, and rapid mass production potential, mRNA vaccines have become an attractive alternative to traditional vaccines and have broad application prospects. The Food and Drug Administration (FDA, the United States) has provided emergency authorization for the use of Moderna’s vaccine for people aged 18 and older. Together with the previously approved Pfizer, New York, USA -Biotech, Mainz, Germany Vaccine, the United States now has two vaccines for emergency use.

Moderna, Cambridge, US and the National Institute of Allergy and Infectious Diseases, NIAID, Massachusetts, USA have collaborated to develop an mRNA-based vaccine (mRNA-1273) consisting of sequence-optimized mRNAs that encode the spike protein encapsulated in lipid nanoparticles. Studies in non-human primates have shown that the vaccine has immunogenicity and protective effects after administration, with animals in the mRNA-1273 group showing only mild inflammation, and no viral RNA or antigens were detected in the lungs [39]. Moderna completed the production of its first new crown vaccine for clinical trials on 7 February 2020. On 4 March 2020, the FDA approved the vaccine’s application for clinical trials. Subsequently, Moderna started Phase I, Phase II, and Phase III clinical trials in March, May, and July 2020, respectively. In phase I (NCT04283461) dose-escalation trials, the vaccine-induced spike protein binding and virus-neutralizing antibody responses in recipients aged 18 to 55 years showed higher antibody titers following the second vaccination [40]. On 16 November 2020, Moderna announced that the first interim analysis of phase III clinical trial (NCT04470427) had reached the primary efficacy endpoint, and the vaccine efficacy reached 94.1% [41]. However, the vaccine needs to be stored at −20 °C, which is a problem for vaccine promotion.

The other mRNA vaccine is the nucleoside-modified mRNA vaccine encapsulated in lipid nanoparticles, developed by BioTech, Mainz, Germany/Pfizer, New York, USA, including the IgG and neutralizing antibody binding to RBD with fewer side effects. The vaccine comprises BNT162b1 and BNT162b2 vaccines, wherein BNT162b1 encodes the SARS-CoV-2 receptor binding domain, and BNT162b2 encodes the SARS-CoV-2 Spike protein. In phase I clinical trials (NCT04368728), both candidate vaccines induced similar dose-dependent SARS-CoV-2 and geometric mean titers (GMTs), with a lower incidence and severity of systemic reactions from BNT162b1 than from BNT162b2, especially in older adults [42]. On November 9, Pfizer announced that the effectiveness of vaccine candidate BNT162b2 to prevent participant COVID-19 exceeded 90% [43]. New research shows that people who have previously been infected with SARS-CoV-2 only need a single dose of BNT162b2 to reach the peak antibody response (6347 U/mL) 7 days after vaccination [44]; anti-RBD IgG, anti-S1/S2 IgG, and total Ig anti-RBD showed correlation (r = 0.99) [45]. These data would help select major drug candidates and dose levels for a large global Phase IIb/III safety and efficacy study to begin as early as July 2020. The companies expected to produce 100 million doses by the end of 2020 and possibly 1.2 billion doses by the end of 2021. As of December 2020, BNT162b2 has obtained emergency use authorization in the United States, Mexico, Canada, and other places, and temporary emergency use authorization in the United Kingdom [46]. BNT162b2 needs to be stored at −80 °C, which may cause major problems for vaccine storage and transportation.

The mRNA-based vaccine (CVnCoV) developed by CureVac AG, Standort Tübingen, Germany, comprised LNP-formulated, non-chemically modified, sequence engineered mRNA encoding a full-length spike protein with two proline mutations (S-2P). The results of preclinical studies showed that immunization with CVnCoV could induce strong humoral responses and robust T-cell responses, and CVnCoV could effectively resist the D614G mutation virus [47]. The phase I trial (NCT04449276) and phase IIa trials (NCT04515147) are ongoing, expected to be completed by the end of 2021. The clinical trial of CVnCoV is currently in phase IIb/III.

### 2.5. SARS-CoV-2 Subunit Vaccines

The subunit vaccine (NVX-CoV2373) developed by Novavax, Gaithersburg, MD, USA is composed of wild SARS-CoV-2 (GenBank accession number: MN908947; nucleotides 21,563–25,384) and Matrix-M1 adjuvant. Phase I/II clinical trials (NCT03615911) began in May 2020, and Phase I results showed that the vaccine was effective in producing antibodies in subjects, with no serious adverse reactions. The levels of neutralizing antibodies produced in the volunteers co-inoculated with the adjuvant were four to six times higher than those in the COVID-19 survivors, proving effective against viral infection. Adjuvant added to a vaccine can obviously improve the effect of vaccination. Vaccines and adjuvants are stored at 2–8 °C, which is advantageous to the BioTech/Pfizer and Moderna mRNA vaccines in terms of storage and transportation temperature [48]. The phase III clinical trial results are expected in early 2021.

CoV2 preS dTM is a recombinant protein vaccine developed by Sanofi, Paris, France/GSK, Brentford, UK which contains the SARS-CoV-2 pre-fusion spike protein. The preS dTM was produced using the Sanofi proprietary insect-cell baculovirus expression vector system. In the phase I/II trial (NCT04537208), participants were vaccinated with 1.3 μg or 2.6 μg of vaccine, respectively, and were given AF03 or AS03 adjuvants in the same dose group, while some participants in the 2.6 dose group were vaccinated without adjuvant. The results showed that the injection response rate of the second dose was higher than that of the first dose, and the injection response rates of the AF03-adjuvanted group, low-dose group, and without-adjuvant group were lower than other groups. After receiving the second dose of the vaccine, NAb could be detected in the serum of participants. The NAb titer of participants over 50 years old was lower than in those who were 18–49 years old. The NAb titer of the low-dose group was lower than that of the high-dose group. The NAb titer of the AF03-adjuvant group was lower than that of the AS03-adjuvant group, and the NAb titer of the high dose group without adjuvant was similar to that of the control group [49]. The dose and formulation of the vaccine will be adjusted in Phase III trials (PACTR202011523101903).

## 3. Clinical Trials on the Ability of Different Types of MERS-CoV Vaccines to Induce Neutralizing Antibody Levels after Vaccination

### 3.1. MERS-CoV Vector Vaccines

Participants in the Chadox1 MERS phase I clinical trial (NCT03399578) received high, medium, and low doses of the vaccine. Chadox1 MERS contains the Chadox1 vector, which expresses a codon-optimized coding sequence for the full-length spike proteins (S1 and S2 subunits) of the MERS-CoV spike proteins. Participants in the phase I clinical trial were followed up for 12 months, and the T-cell response was sustained up to 1 year after vaccination. The seroconversion rates were 20% in the low-dose group, 0% in the medium-dose group, and 20% in the high-dose group. Chadox1 MERS could thus induce cellular and humoral immunity. In the pseudovirus neutralization experiment, 71–79% of participants produced neutralizing antibodies. The results of phase I clinical trials of the vaccine supported entry into phase Ib and phase II trials [50]. MVA-MERS-S is a vaccine using the vaccinia virus as the carrier, which encodes the complete MERS-CoV spike protein. The phase I trial (NCT03615911) used two doses of 1 × 10^7^ PFU and 1 × 10^8^ PFU with a booster dose on day 28. After the booster dose, seroconversion was detected in 75% (9/12) of the low-dose group and 100% (100/100) of the high-dose group, and the induced humoral and cellular immunity was dose-dependent. Antibody response peaked on days 42 and 56 and continued until day 84. Six months after vaccination, most participants’ antibody responses declined to baseline levels [51]. The experience and results of this trial have been used to guide the development of a SARS-CoV-2 vaccine.

### 3.2. MERS-CoV DNA Vaccines

The MERS-CoV DNA vaccine (GLS-5300/INO-4700) was the first MERS vaccine to enter clinical trials (NCT02670187) (Table 2). It contains a 6 mg/mL plasmid pGX9101. The plasmid contains the optimized, full-length, microconsensus of the MERS spike protein. The phase I trial was conducted using three doses of GLS-5300/INO-4700 after intramuscular injection, and three injections per dose. Neutralizing antibodies were detected in 34 of the 68 participants, and the half-to-maximum neutralizing GMT peaked at week 14 in a range of 7.9 to 508. The vaccine-induced cellular immune and antibody responses were similar to those in convalescent patients. A dose-independent immune response was detected in 85% of participants after two inoculations, and this immune response persisted for one year in 3% of participants [52]. Phase I/II clinical trial (NCT03721718) is underway in Korea.

## 4. Clinical Trials on the Ability of Different Types of SARS-CoV Vaccines to Induce Neutralizing Antibody Levels after Vaccination

### 4.1. SARS-CoV DNA Vaccines

VRC is a DNA vaccine comprising a recombinant plasmid encoding the SARS spike protein. The plasmid can clone a single protein into the expression vector CMV/R, which has been used in DNA vaccines against HIV and Ebola. The clinical trial used the Biojector 2000^®^ Needleless Injection Management System™ to administer three doses of 4 mg each. No neutralizing antibody was detected in the microneutralization plaque-reduction neutralization assay (MNA). Pseudoverbal neutralization tests detected seroconversion in 80% of participants, which peaked at 8 to 12 weeks, and 60% of participants remained positive at 32 weeks [53].

### 4.2. SARS-CoV Inactivated Vaccines

ISCV was prepared by inactivation of the SARS-CoV Sino3 strain with β-propanolactone. It was divided into two dose groups: 16SU and 32SU (Table 3). The serum of both groups was positive on the 42nd day. On day 56, most of the participants (23/24) maintained positive serum antibodies, but the average antibody titer was lower than that on day 42. The GMT of the recovery period was 61 U. Using the same laboratory standards and methods, the GMT values of the low-dose group and the high-dose group on day 42 were 31.1 U and 30.9 U, respectively, half of that of the patients in the convalescence period of SARS. Further trials are needed to establish the criteria for evaluating the efficacy of SARS vaccines and to optimize vaccine doses and timetables. Epidemiological data from 2003 showed that no re-infection of SARS was reported during the eight-month global epidemic [54].

## 5. Methods to Detect Neutralizing Antibodies

PRNT, microneutralization assay (MNA), fluorescent neutralization assay (FNA), and pseudovirion neutralization assay (PsVNA) are the main methods to detect the neutralizing antibody titer. Before all neutralizing antibody titer tests, serum samples should be incubated at 56 °C for 30 min to inactivate the complement, and the test should be started after balancing to room temperature.

### 5.1. Plaque Reduction Neutralization Test, PRNT

PRNT is the current laboratory standard for the determination of neutralizing antibodies, and needs to be detected with live viruses. The research found that PRNT IC_90_ is more sensitive than MNA in detecting antibodies [55]. However, PRNT has low throughput, so it is not suitable for large-scale serodiagnosis and vaccine evaluation. The diluted serum is mixed with the same volume of virus suspension. After incubation at 37 °C for one hour, the mixture is added to the monolayer Vero E6 cells. Adsorption at 37 °C for one hour is performed, and then the cells are covered with DMEM containing 2% high gel temperature agar, 5% FBS, and 1% P/S [56]. After incubation at 37 °C for 3–5 days, the plaques are fixed with formaldehyde, stained with crystal violet, and counted under a microscope. Alternatively, HRP-labeled secondary antibodies are used, and the number of infected cells per well is calculated using an ImmunoSpot^®^ Image analyzer. The 50% plaque reduction (IC_50_) value is often used. The neutralization titer is calculated as the reciprocal of the highest serum dilution that reduces plaque formation by 50%. In addition, IC_80_ (80% plaque reduction) and IC_90_ (90% plaque reduction) are also calculated. The use of appropriate reference materials to standardize procedures might lead to a better inter-laboratory comparison of results after each laboratory’s validation.

### 5.2. Fluorescent Neutralization Assay, FNA

FNA can detect neutralization antibodies and produce the same results as that of the PRNT (the gold standard for serological tests) [56]. However, the detection time of FNA is several days less than that of PRNT, which makes it a high-throughput method to detect neutralizing antibodies [56]. The mNeonGreen gene is inserted into the virus strain used in the experiment, which could make the cells turn into fluorescent cells after infection. The diluted serum is mixed with an equal volume of virus suspension. After incubation at 37 °C for one hour, the monolayer cells are inoculated with the mixture at a multiplicity of infection (MOI) of 0.5. After 16 h, the number of fluorescent cells is detected. By plotting the dose-response curve between the number of fluorescent cells and the dilution ratio of the serum, the dilution ratio that neutralized 50% of the fluorescent cells (NT_50_) could be determined, which gives the neutralizing antibody titer.

### 5.3. Pseudovirus Neutralization, PsVNA

PsVNA is a neutralization test using pseudovirus instead of living viruses, which is safer and more sensitive than PRNT. SARS-CoV-2 pseudovirus particles include human immunodeficiency virus (HIV)-based lentiviral particles [57], murine leukemia virus (MLV)-based retroviral particles [58], and vesicular stomatitis virus (VCV) [59], which all contain the spike protein of SARS-CoV-2. PsVNA is safer than other neutralization tests using live viruses, because the live virus neutralization test of SARS-CoV-2, MERS-CoV, and SARS-CoV must be carried out in a biosafety level 3 (BSL-3) laboratory, which is a great limitation for neutralizing antibody detection. Different strains are usually used in live virus neutralization tests in different laboratories, which is not conducive to the comparison of results between laboratories; by contrast, the results of PsVNA were more convenient to compare and had a good correlation with the live virus neutralization test [60]. The HEK293 cell line transfected with ACE2 [23], or HEK293T cell line transfected with ACE2 and TMPRSS2 [61], are added to the wells after incubation at 37 °C for 1 h. After incubation at 37 °C for three days, the cells are lysed, the activity of luciferase is detected, and the inhibition rate is calculated. The ID_50_ is calculated for the test well, which was defined as 50% inhibition of the fluorescence compared with the positive control well, and the neutralizing antibody titer was the reciprocal of the highest serum dilution. ID_80_ was defined as the test well in which 80% of the luciferase activity was inhibited, compared with the positive control well.

## 6. Conclusions

Vaccines are one of the most important means to prevent and control infectious diseases. Since the global outbreak of COVID-19, countries around the world have accelerated the development of SARS-CoV-2 vaccines. Although the current phase III clinical trial results show that the vaccines have good safety and immunogenicity, and the approved vaccines have good protective effects, the long-term protective effects and adverse reactions of the vaccines still need to be tested to better respond to the new COVID-19 epidemic.

Vaccines for SARS-CoV-2, MERS-CoV, and SARS-CoV include inactivated vaccines, vector vaccines, DNA vaccines, mRNA vaccines, and protein subunit vaccines. These different types of vaccines have their advantages. Among them, as a first-generation vaccine technology, inactivated viruses have already been marketed as a variety of vaccines based on this mature technology. Similar phenomena to ADE have been observed in the clinical trials of MERS-CoV and SARS-CoV vaccines [50,51,52,53,54], while the clinical trials of SARS-CoV-2 vaccines have not reported ADE.

In the course of using the SARS-CoV-2 vaccines, the recipients had serious adverse reactions such as thrombosis, which even led to death. This adverse reaction occurred in the use of vaccines using chimpanzee adenovirus (University of Oxford/AstraZeneca) and adenovirus type 26 (Janssen/Johnson & Johnson) as vectors. It is worth noting that these two candidate vaccines are vector vaccines. No cases of CVST with thrombocytopenia have been reported after receipt of either of the two mRNA COVID-19 vaccines authorized for use in the United States. With the widespread use of the SARS-CoV-2 vaccine worldwide [62], serious adverse events such as those that may lead to death should be paid attention to. The existing reported cases indicate the importance of rapid identification of this rare disease [24].

The storage and transportation conditions of the vaccine also play a vital role in the application of the vaccine. Inactivated vaccines can be stored at 2–8 °C because of their good thermal stability, followed by adenovirus vaccines that can be stored at 4 °C or as freeze-dried powders, protein subunit vaccines that can be stored at 2–8 °C for six months, and DNA vaccines that can be stored at room temperature for one year. mRNA vaccines have the most demanding storage and transportation requirements, one of which needs to be stored at −80 °C, and the other needs to be stored at −20 °C.

Some vaccine companies have also conducted clinical trials of the SARS-CoV-2 vaccine in patients; for example, one study vaccinated 658 patients after solid organ transplantation with two doses of mRNA vaccine, but only 15% of patients had an antibody response after receiving two doses of vaccine [63]. It may be that the immunosuppressive drugs used after transplantation affect the humoral response.

Recently, the SARS-CoV-2 variants that appeared in Britain, South Africa, Brazil, India, and other places have become a hot topic; we need to focus on whether the protective effect of vaccines is effective for the mutations. The new variants, such as B.1.1.7 (501Y.V1) in the United Kingdom, B.1.351 (501Y.V2) in South Africa, and B.1.1.28.1 (P.1) in Brazil mark the beginning of the antigenic drift for SARS-CoV-2. Mutations may lead to the failure of antiviral drugs, diagnostic test failures, and resistance against antibodies. The neutralizing antibodies target the RBD domain motifs of spike protein, and some mutations might impact binding and neutralization [64]. The mRNA-1273 (Moderna) vaccine could be effective against the N501Y modification, but the effectiveness of mRNA-1273 for the 501Y.V2 variant and India variant is unclear. Research on BNT162b2 found that 501Y.V1 (69/70-deletion + N501Y  +  D614G) and 501Y.V2 (E484K  +  N501Y  +  D614G) could be effectively neutralized by the serum elicited by two doses of BNT162b2 [65].

SARS-CoV-2 vaccine candidates targeting the spike protein sequence (original D614G ancestral) single epitope might have no efficient immune response to the new variants [21]. The easy mutation of single-stranded positive-stranded RNA viruses is a hidden danger of current vaccine candidates. Therefore, vaccine companies should formulate a response plan for the mutant SARS-CoV-2 strains promptly to prevent a repeated epidemic of COVID-19.

With the joint efforts of countries around the world, some of the COVID-19 vaccines have completed phase III clinical trials in less than a year and have even been authorized for emergency vaccination or conditional market use by the population. The development of these vaccines will also provide experience for future vaccine development and the emergency development of vaccines for the prevention and control of other diseases. A neutralizing antibody is a type of antibody produced by B lymphocytes that can bind to the antigen of the virus, thereby preventing the virus from adhering to the target cell receptor and preventing cell invasion. Neutralizing antibodies can destroy the virus before the virus enters the cell; therefore, the presence of neutralizing antibodies in the body in distress can prevent infection from the corresponding virus. This study analyzed the levels of neutralizing antibodies produced by different types of vaccines in the human body to provide systematic analysis and data support for the future development of more effective and safer vaccines.

## Figures and Tables

**Figure 1 vaccines-09-00588-f001:**
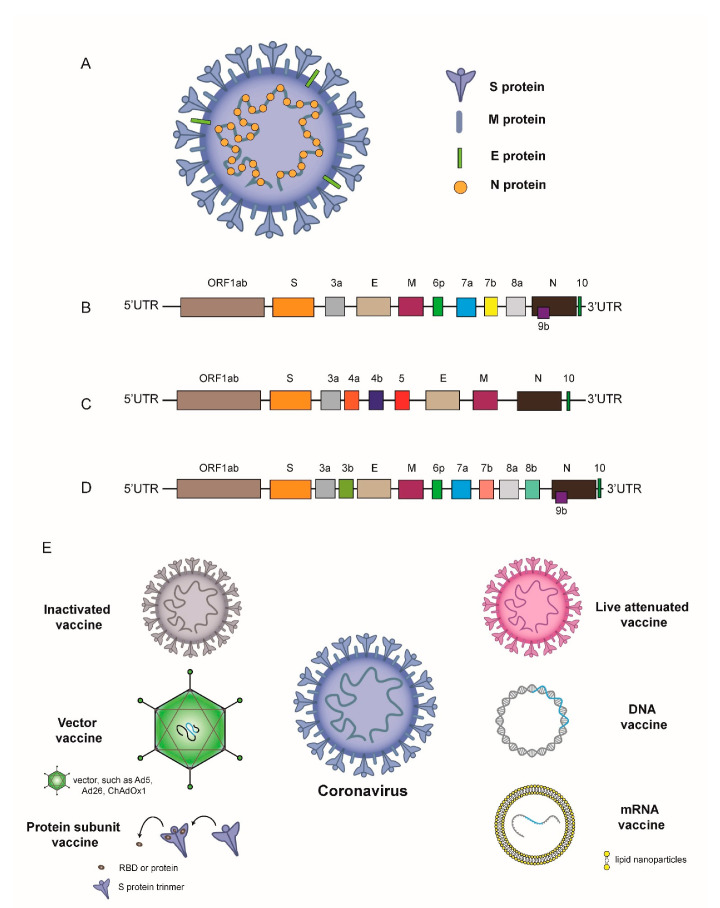
(**A**) Structure of coronavirus; (**B**–**D**) the schematic genomic structure of coronaviruses: (**B**) SARS-CoV-2; (**C**) MERS-CoV; (**D**) SARS-CoV; (**E**) Schematic diagram of different types of coronavirus vaccines.

**Table 1 vaccines-09-00588-t001:** Clinical trial results of SARS-CoV-2 candidate vaccines.

Developer	Vaccine Type	The Vaccine Dose	The Booster Dose	Number of Doses	Neutralizing Antibody GMT	The Control Group
Sinovac [14]	Inactivated vaccine (CoronaVac)	3 μg, 6 μg	Day 14 or Day 28	2	Day 42, MNA IC_50_ were 23.8 in the 3 μg group and 30.1 in the 6 μg group; Day 56, MNA IC_50_ were 44.1 in the 3 μg group, and 65.4 in the 6 μg group.	MNA IC_50_ were 163.7 (convalescent asymptomatic patients) and 76.1 (hospitalized patients)
Wuhan Institute of Biological Products/Sinopharm [16]	Inactivated vaccine	5 μg	Day 14 or Day 21	2	14 days after the whole injection, PRNT IC_50_ were 121 on day 0 and 14 group, and 247 on day 0 and 21 group.	/
Beijing Institute of Biological Products/Sinopharm [17]	Inactivated vaccine (BBIBP-CorV)	4 μg, 8 μg	Day 14 or Day21 or Day 28 (except the 8 μg group)	2 or 1	28 days after the injection, in the 4 μg group, MNA IC_50_ were 169.5 on day 0 and 14; 282.7 on day 0 and 21; and 218.0 on day 0 and 28; 14.7 in the 8 μg group.	/
Bharat Biotech [20]	Inactivated vaccine (BBV152)	3 μg, 6 μg	Day 28	2	Day 56, PRNT IC_50_ were 100.9 in the 3 μg group and 197.0 in the 6 μg group; MNA IC_50_ were 92.5 in the 3 μg group and 160.1 in the 6 μg group.	/
CanSino Biological/Beijing Institute of Biotechnology [27]	Adenovirus Type 5 Vector vaccine	5 × 10^10^, 1 × 10^11^ Virus particles	/	1	Day 28, LTIA [27] were 19.5 in the high dose group and 18.3 in the low dose group; PBNA EC_50_ was 61.4 in the high dose group and 55.3 in the low dose group.	/
Janssen/Johnson & Johnson [29]	Adenovirus Type 26 Vector vaccine (Ad26.COV2.S)	5 × 10^10^, 1 × 10^11^ Virus particles	Day 28 (a part of every dose group)	2 or 1	Day 29, MNA IC_50_ were 224 (5 × 10^10^ VP, one dose), 224 (5 × 10^10^ VP, two-dose), 215 (1 × 10^11^ VP, one dose), 354 (1 × 10^11^ VP, two-dose). Day 56, MNA IC_50_ were 310 (5 × 10^10^ VP, one dose), 288 (5 × 10^10^ VP, two-dose), 370 (1 × 10^11^ VP, one dose), 488 (1 × 10^11^ VP, two-dose)	/
University of Oxford/AstraZeneca [23]	Vector vaccine (ChAdOx1 nCoV-19)	5 × 10^10^ Virus particles	Day 28 (10 volunteers received)	2 or 1	Day 28, PRNT IC_50_ was 218, MNA IC_80_ was 51, and VN IC_100_ was 29; the MNA IC_80_ of receiving booster dose was 136.	/
Inovio [37]	DNA vaccine (INO-4800)	1.0 mg, 2.0 mg	Week 4	2	Week 6, PRNT IC_50_ were 102.3 in the 1.0 mg group and 63.5 in the 2.0 mg group.	PRNT IC_50_: 160
Moderna/NIAID [40]	mRNA vaccine (mRNA-1273)	25 μg, 100 μg, 250 μg	Day 28	2	Day 43, PsVNA ID_50_ were 112.3 in the 25 μg group, 343.8 in the 100 μg group, 332.2 in the 250 μg group. PRNT IC_80_ were 339.7 in the 25 μg group, 654.3 in the 100 μg group, and 332.2 in the 250 μg group.	PsVNA ID_50_: 109.2, PRNT IC_80_: 158.3
BioNTech/Pfizer [42]	mRNA vaccine (BNT162b1)	10 μg, 30 μg, 100 μg	Day 21 (except the 100 μg group)	2 or 1	Day 28, FNA IC_50_ were 168 in the 10 μg group and 267 in the 30 μg group. Day 35, FNA IC_50_ were 180 in the 10 μg group, 437 in the 30 μg group; Day 21, FNA IC_50_ was 33 in the 100 μg group.	FNA IC_50_: 94
Novavax [48]	Protein Subunit (NVX-CoV2373)	5 μg, 25 μg (with or without adjuvant)	Day 21	2	Day 35, MN IC_>99%_ were 3906 in the 5 μg group (with adjuvant), 3305 in 25 μg group (with adjuvant), 41 in 25 μg group (without adjuvant), and 128 in 25 μg group (with adjuvant in the first dose and without adjuvant in the second dose).	MN IC_>99%_ were 254 (asymptomatic patients) and 837 (symptomatic patients)
Sanofi Pesteur/GSK [49]	Protein Subunit (CoV2 preS dTM)	1.3 μg (with adjuvant), 2.6 μg (with adjuvant), 2.6 μg (without adjuvant)	Day 22 (a part of per dose group)	2 or 1	Day 36, MNA IC_50_ were 13.1 in the 1.3 μg group (AF03-adjuvant), 20.5 in the 1.3 μg group (AS03-adjuvant), 43.2 in the 2.6 μg group (AF03-adjuvant), and 75.1 in the 2.6 μg group (AS03-adjuvant). NAb titer was not deteced in the 2.6 μg group (wtihout adjuvant).	/

No special indication means all dose groups received the booster dose. The control group: the neutralizing antibody level of convalescent patients with COVID-19; GMT: geometric mean titer; PRNT: plaque reduction neutralization test; MNA: microneutralization assay; FNA: fluorescent neutralization assay; PsVNA: Pseudovirus neutralization; VP: Virus particles; VN: live virus neutralisation.

**Table 2 vaccines-09-00588-t002:** Clinical trial results of MERS-CoV candidate vaccines.

Developer	Vaccine Type	The Vaccine Dose	The Booster Dose	Number of Doses	Neutralizing Antibody GMT	The Control Group
University of Oxford/AstraZeneca [50]	Vector vaccine (ChAdOx1 MERS)	5 × 10^9^ VP, 2.5 × 10^10^ VP, 5 × 10^10^ VP	/	1	PsVNA ID_50_ was 12.5 in the low dose group, 25 in the middle dose group, and 50 in the high dose group	/
German Center for Infection Research [51]	Vaccinia virus Ankara vector vaccine	1 × 10^7^ PFU, 1 × 10^8^ PFU	Week 4, Week 12	3	Week 14, MNA NT_50_ was 10	MNA NT_50_ was 70 in the acute phase and 40 in the convalescent phase
GeneOne Life Science/Inovio [52]	DNA vaccine (GLS-5300/INO-4700)	0.67 mg, 2 mg, 6 mg	Day 28	2	PRNT IC_80_ was positive in 75% of the low-dose group and 82% of the high-dose group	/

No special indication means all dose groups received the booster dose. The control group: the neutralizing antibody level of convalescent patients with MERS; GMT: geometric mean titer; PRNT: plaque reduction neutralization test; MNA: microneutralization assay; PsVNA: pseudovirus neutralization; VP: virus particles; PFU: plaque-forming unit.

**Table 3 vaccines-09-00588-t003:** Clinical trial results of SARS-CoV candidate vaccines.

Developer	Vaccine Type	The Vaccine Dose	The Booster Dose	Number of Doses	Neutralizing Antibody GMT	The Control Group
NIAID [53]	DNA vaccine (VRC)	4 mg	Day 28, Day 56	3	No neutralizing antibodies were detected in PRNT IC_80_, neutralizing antibodies detected in PsVNA ID_50_ peaked at 8 to 12 weeks	/
Sinovac Biotech [54]	Inactivated vaccines (ISCV)	16 SU, 32 SU	Day 28	2	The neutralizing antibody peaked at 2 weeks after the second immunization and decreased at 4 weeks after immunization	/

No special indication means all dose groups received the booster dose. The control group: the neutralizing antibody level of convalescent patients with SARS; GMT: geometric mean titer; PRNT: plaque reduction neutralization test; PsVNA: pseudovirus neutralization; SU: SARS-CoV units.

## Data Availability

Not applicable.

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
