# Peer review of "Comparison and Analysis of Neutralizing Antibody Levels in Serum after Inoculating with SARS-CoV-2, MERS-CoV, or SARS-CoV Vaccines in Humans"

_vaccines, 2021, doi:10.3390/vaccines9060588_

Round 1

Reviewer 1 Report

In this manuscript, Yu et al compared and analyzed serum neutralizing Ab levels after inoculating SARS-Cov2, MERS-Cov or SARS-Cov vaccines in humans. This paper is well-written, but need to be very strongly updated until april 2021 regarding the fast-evolving field of SARs-Cov2 vaccines, new results of pre-clinical and clinical trials, new papers on mid- and long-lasting Ab neutralizing responses.

For example, numerous studies, describing Ab responses, have been published on Pfizer, Moderna, Astra/Zeneca and JJ/Janssen vaccines (and others), with large cohorts of vaccinated patients, that need to be updated. CureVac mRNA vaccine, and Sanofi/GSK subunit vaccine need to be integrated, discussed in the review (and also mentioned in Table 1). New results on Inovio DNA vaccine have been now announced by the company.

Please correct a mistake in Table 1 in front of INO-4800, the company being Inovio and not Moderna. Others companies involved in DNA technology have to be mentioned and updated in Table 1. Do not forget to discuss about toxicities (ie. Thrombosis for Astra/Zeneca or Janssen vaccines), about SARS-Cov2 variants and their implication/consequences in Ab neutralization.

Minor point : indicate legend of Figure 1 just under the figure at the bottom of pg2. In front of these considerations of strong updating, new questions, new papers not mentioned, I cannot recommend this paper for publication in this present form.

Author Response

Dear reviewer,

Thank you for your valuable comments and suggestions on this manuscript. We made some changes to the manuscript.

Kind regards,

Reviewer 2 Report

The present Review is interesting but there are many summary points about this argument published yet and in general for many aspects of the SARS-CoV-2 vaccine program.

So, I think that is better for the Authors to improve (seriously) the Manuscript.

Point1. Update the statistics for SARS-CoV-2 and MERS infected, deaths etc.

Point2. Re-think Figure 1, the article needs a Figure with a schematization of different type of Vaccines with the elements that characterize these strategies and vaccine particles. In other words a more complex Figure, with a detailed legend is necessary. The current Figure is too simple and not appropriate for a journal of this level.

Point3.  Authors should organize better the Tables reducing the font size to have a better disposition of the words inside the columns. Not all the data in Tables are reported correctly please read again the Tables (for example amount of the second dose for Sinopharm is not present).

Point4. When Authors present the vaccines and in particular adenoviral vector based vaccines is necessary to specifiy the type of adenovirus adopted. Example: ape adenovirus is wrong but chimpanzee is more correct and what about the different human adenovirus used?

Point5. The data reported are focalized only about trial but are not complete in their exposition. The problem is that the title of the manuscript and part of the text settings would suggest a more extensive and broader discussion on vaccines. This is a critical point; the manuscript should be improved if you think only about trials with a new Title or extended twice all the manuscript if you choose to describe more aspects about coronavirus vaccines. 

Point6. This is another critical point. A complete examination about Coronavirus vaccines need to be updated at the moment of the first run of review process (today). In your manuscript the Janssen J&J vaccine is poorly discussed the Indian vaccine Bharat is not discussed and mentioned, other trials are not reported. Please reconsider the manuscript in a more complete format because there are lot of informations in world wide web and there are many websites useful about coronavirus vaccines. This is a scientific paper that needs to be more interesting than a simple collection of informations.

Author Response

(The authors gave the same response as above.)

Reviewer 3 Report

In the review manuscript entitled “Comparison and analysis of serum neutralizing antibody levels after inoculating SARS-CoV-2, MERS-CoV, or SARS-CoV vaccines in humans” the authors described different vaccine formats that are being pursued in the clinical development for SARS-CoV, MERS-CoV, and SARS-CoV-2  viruses and summarized a comparative analysis of neutralizing antibodies in the sera of patients infected with these coronaviruses. The review will be a nice addition to other reviews and literature in the field of coronaviruses. Minor suggestions that could improve the broad readership of this review.

It would be great if the authors could include a brief background about the biology and biochemistry of these viruses. This will help better understand the comparison made by the authors.

It would be idle if the authors could include the clinical trial ids in the table to direct the readers to appropriate reference if they wish to learn additional details.

Citations need to be appropriately included. For eg. work mentioned in the section “Methods to detect neutralizing antibodies” has not been referenced.

Author Response

(The authors gave the same response as above.)

Reviewer 4 Report

The authors reviewed current SARS-CoV-1, -2 and MERS-CoV vaccine technologies that received emergency use authorization or undergoing clinical investigation. The manuscript is generally readable. However, the authors used inconsistent and difficult terminology at several instances that alters the knowledge they wanted to convey. Some examples are given below

  1. line 78-79, I would suggest the sentence along the lines "specific vaccine formulations may be required in different populations....
  2. In the figure 1, mRNA liposome encapsulation is missing
  3. footnotes of table missing
  4. Line 87, influenza vaccines equally comprise both inactivated and subunit vaccines. The authors indicated most are inactivated. Please explain
  5. Use consistent terminology as shown here: 1. Sero conversion (several terms were used in the manuscript such as serum conversion, positive conversion etc) 2. full length spike glycoprotein, S1/2 or RBD as appropriate (several terms were used in the manuscript such as synaptic glycoprotein,spikelet etc)
  6. The term "enhancing immunity" (line 138; lines 255-256) is used at several places. I would recommend the authors to use a more relevant term to convey the meaning
  7. line 184-185 the time periods were past. Please explain the results of this clinical trial
  8. line 215, "mRNA vaccine prepared from lipid nanoparticles...." Please correct the sentence along the lines mRNA encapsulated in lipid nanoparticles
  9. line 229, storage and transportation
  10. lines 235-236, Please re-frame the sentence. It sounds like volunteers were inoculated with just adjuvant. Use term 'coinoculated'
  11. line 239 rephrase the term 'superior'. advantageous?
  12. line 246 codon optimized sequence of the full length..... correction needed
  13. line 248-252. rephrase the sentences to convey exact meaning
  14. lines 265-266, plasmid inserted into..... Sentence require rephrasing
  15. lines 284-285, the original study used MNA whereas authors indicated PRNT. Correction required
  16. The authors did not indicate the use of agar in performing PRNT
  17. Section 5, mere review of neutralizing antibody detection methods doesn't really enhance one's knowledge. I would recommend the authors to include advantages and disadvantages of these methods in rapid detection of neutralizing antibodies.
  18. Sub-section 5.5, lines 357-358, Please also indicate that HEK293T-ACE2/TMPRSS2 cells are also used to conduct PsVNA. Reference studies PMID: 33690649 and PMID: 32485970
  19. Lines 377-384 references missing and several general erroneous claims have been made. Please rephrase the sentences.
  20. lines 408-411, this statement is wrong. several existing and emerging variants and mutants except D614G and B.1.1.7 are known to escape neutralizing antibodies. rephrasing needed.

Author Response

(The authors gave the same response as above.)

Round 2

Reviewer 1 Report

Corrections requested have been done.

Author Response

(The authors gave the same response as above.)

Reviewer 2 Report

The manuscript is really more intersting in the present form.

Read to fix minor language and editing problems like line 155 parenthesis

or center in the page the Figure 1 please see panels B, C, and D.

Author Response

(The authors gave the same response as above.)

Reviewer 4 Report

The manuscript is improved in the revised version. However, I recommend authors to carefully make grammatical and English language corrections before publication.

Minor comments

  1. Add "RNA" next to the word stranded on line 30
  2. Line 33, ORFs need to put before protein identifiers. ORF3a, 3b etc
  3. Please add reference PMID: 32485970 in the first paragraph of introduction near accessory proteins

Author Response

(The authors gave the same response as above.)
